# Potential Applications of Food-Waste-Based Anaerobic Digestate for Sustainable Crop Production Practice

**Jonathan Ries, Zhihao Chen and Yujin Park ***

College of Integrative Sciences and Arts, Arizona State University, Mesa, AZ 85212, USA; jries5@asu.edu (J.R.);
zhihao.chen.2@asu.edu (Z.C.)
* Correspondence: yujin.park.2@asu.edu

**Abstract:** The global food system is and will be facing many challenges in the coming decades, which will necessitate innovative solutions to address the issues of a diminishing fertilizer supply, an increasing food demand from growing populations, and frequent extreme climates due to greenhouse gas emissions. An advancement proposed is the synthesizing of fertilizer from food waste, here referred to as food waste anaerobic digestate (FWAD). This occurs through the process of anaerobic digestion, where organic matter such as food waste is contained in an anaerobic environment and allowed to be broken down by microorganisms. One of the resulting products is anaerobic digestate, which possesses the necessary nutrients for effective fertilization for crop production. In addition to reducing greenhouse gases and waste in landfills, the replacement of synthetic fertilizers with ones made from food waste would help to alleviate the impacts of the current fertilizer shortage being experienced worldwide, which will be exacerbated by a reducing supply of materials needed for synthetic fertilizer production. In this paper, we discuss the nutrient characteristics of FWAD, and recent studies utilizing FWAD in horticulture and crop production, to advance our understanding of the effectiveness and challenges of using FWAD as a fertilizer. By employing appropriate application methods, such as nitrification, dilution, and amendment, FWAD demonstrates considerable potential as an effective fertilizer for a wide range of leafy greens and some fruiting crops.

**Keywords:** anaerobic digestates; bio fertilizer; circular economy; food waste; organic fertilizers

## 1. Introduction

Excess food waste has been an increasingly difficult problem to manage, taking up the majority of landfill space in the U.S. and resulting in 30–40% of the total U.S. food supply being lost to waste [1]. A total of 40 million tons of food waste is discarded every year in the U.S. alone, and 1.4 billion tons of food waste is discarded each year worldwide [1], which contributes 6–10% of the total greenhouse gas emissions by humans [2]. A practical use for food waste has been developed which involves breaking down the waste into simpler components through a process called anaerobic digestion (AD). This process produces digestate and biogas, each of which has useful components, including ones that can be used as a feasible fertilizer in the form of food waste anaerobic digestate (FWAD). In 2018, 56% of all non-industrial food waste in the U.S. was landfilled, with merely 8% being used to produce digestate [3]. Besides the digestate product that is produced from the AD process, biogas in the form of methane ($CH_4$) and carbon dioxide ($CO_2$) is produced as well, which can be used as an energy source for electricity and heat, reducing greenhouse gas emissions [4]. Using these waste products as inputs into energy and food systems represents a shift from a linear model, where waste products go unused, to a circular model, where waste can be reintroduced into the food system and used in an efficient manner. Thus, the production and use of fertilizers derived from food waste in a "circular food system" is starting to be recognized in terms of sustainable agriculture [5]. Recently, many studies have gradually revealed the benefits and challenges of using FWAD as a fertilizer [6].

Synthetic fertilizers have recently been used in greater amounts in agriculture to enhance crop yields. This increased demand has led to a reduction in phosphorus (P) reserves to levels so low that some estimate they may be depleted within 100 years, with peak production taking place in 2030 [7]. In addition, due to recent global events, the supply of materials for synthetic fertilizers has been further diminished, in some cases causing fertilizer prices to double in 2022 [8]. The processes involved with manufacturing synthetic nitrogen fertilizers contribute more than 50% of the greenhouse gas emissions attributed to the agriculture sector, which is responsible for over 10% of total greenhouse gas emissions worldwide [9]. The increasing use of FWAD as a replacement for synthetic fertilizers would alleviate many of these issues related to the production and use of synthetic fertilizer.

Traditionally, the AD process has been used primarily to produce biogas, with digestate being a residual product seen as a bottleneck in the process of biogas production [10]. Digestate products are typically rich in nitrogen (N) and contain most necessary plant mineral nutrients (Table 1), making them usable for plant fertilization [11]. There has been increasing interest in using the digestate, particularly in liquid form, as a fertilizer for use in soilless substrate systems [11,12] and in hydroponics [13–15]. Increased experimentation using FWAD as a fertilizer has led to a better understanding of its effectiveness as such; however, challenges are still associated with its large-scale application. Table 1 shows the typical nutrient requirements by plants, and is compared with three types of FWAD to demonstrate the variation that can be found between different types of FWAD as well as the nutrients that are typically in high or low levels. To ensure the sustainable use of FWAD, nutrient management, pathogen removal, and implementation of standardized practices should be established [16]. This review will analyze the range of potential uses for FWAD as a fertilizer for crop production, report the obstacles associated with its production and use, and compare recent techniques for enhancing FWAD to achieve similar results to those seen with traditional synthetic fertilizers.

**Table 1.** Plant requirements for mineral nutrients and reported nutrient characteristics of food waste anaerobic digestates (FWAD).

| % of Element Relative to N | Typical Requirements by Plants | Nutrient Characteristics of FWAD | | |
|---|---|---|---|---|
| References | [17] | [12] | [18] | [19] |
| N | 100 | 100 | 100 | 100 |
| P | 13–19 | 7 | 6 | 15 |
| K | 45–80 | 22 | 3 | 191 |
| Ca | 5–15 | 8 | 7 | 22 |
| Mg | 5–15 | 2 | 1 | 2 |
| S | 8–9 | – | 0.78 | 8.31 |
| Fe | 0.7 | 2.84 | 0.82 | 9.09 |
| Mn | 0.4 | 0.10 | 0.03 | 0.15 |
| Zn | 0.06 | 0.12 | 0.06 | 0.26 |
| Cu | 0.03 | 0.092 | 0.006 | 0.091 |
| B | 0.2 | – | – | 0 |
| Mo | 0.003 | 0.0017 | 0.0000 | 0.0031 |
| Cl | 0.03 | 21.5 | 42.0 | 50.3 |

## 2. Food Waste Anaerobic Digestate

The AD process uses microorganisms in an anaerobic environment to break down the components of organic matter found in food waste into two major products: biogas and digestate. The biogas produced is in the form of $CH_4$ and $CO_2$, which can be used as an energy source in the same way that natural gas is used [12]. The digestate obtained can be in liquid or solid form, each possessing different properties. The solid portion tends to contain more P and can be used as a substrate or soil amendment [17,20], while the liquid

portion contains more N and potassium (K) and can be used as an effective fertilizer in hydroponic and soilless substrate growing systems [13,14,17].

The food waste used to produce FWAD in developed countries is often sourced from the latter stages of the food supply chain and is composed of uneaten food from schools, homes, restaurants, or cafeterias [21]. The overall composition of this food waste is about 50% cereals, 20% fruits and vegetables, 10% dairy, and 6% meat, with the balance made up of mostly tubers and oils [21]. This contrasts with developing countries, where the bulk of food waste is created during the harvesting process [21]. Food waste generated at all stages of the food supply chain is suitable for use in producing FWAD. It has been reported that just 5% of all wasted food in the U.S is currently being recycled using composting or AD [22]. The lack of currently recycled food waste products indicates there is great potential for available waste to produce FWAD.

The increased production and use of FWAD as a supplement or replacement for synthetic fertilizers can produce many desirable environmental outcomes [16]. The $CH_4$ and $CO_2$ produced during the AD process can be harnessed and used as a source of energy, reducing carbon emissions associated with the disposal and decomposition of food waste in landfills [4]. Global food wastage is estimated to contribute 3.3 gigatons of $CO_2$ equivalent to global carbon emissions [1]. Annually, diverting wasted food from disposal to FWAD production has been estimated to reduce carbon emissions by 190,000 tons of $CO_2$ equivalent in the United Kingdom and by 414,898 tons of $CO_2$ equivalent in Australia [23,24]. Additionally, the effective fertilization of crops using FWAD would reduce the reliance on synthetic fertilizers in the agriculture industry, reducing the greenhouse gas emissions associated with the production and application of synthetic fertilizers [9].

However, it is also essential to consider the potential risks and drawbacks associated with FWAD utilization. The overapplication of FWAD, particularly in the field, can lead to nutrient leaching and eventual runoff into water sources which can cause pollution and eutrophication, a problem exacerbated by the high ammonium ($NH_4^+$) content in FWAD [25]. Pathogens can be present in food waste before the digestion process, but the digestion process has been shown to effectively eliminate or reduce pathogen populations below the required threshold for effective crop fertilization [16]. Furthermore, the large amount of energy required to complete the AD process can offset the reuse of biogas produced during digestion, which necessitates further optimization of the techniques used to produce FWAD and the use of renewable energies whenever feasible [26].

## 3. Nutrient Characteristics of Food Waste Anaerobic Digestate

The nutrient composition of FWAD differs based on the source of the food waste and can change during any treatments applied, but FWAD typically possesses the necessary plant mineral nutrients required for effective plant fertilization in varied proportions (Table 1). FWAD is usually rich in N, chlorine (Cl), and sodium (Na), while showing lower concentrations of P, sulfur (S), boron (B), magnesium (Mg), manganese (Mn), and molybdenum (Mo). Here, the typical ranges of nutrient concentrations in FWAD are reported, and contemporary methods to alleviate nutrient management challenges are discussed.

### 3.1. High $NH_4^+$

The overall percentage of N in FWAD is usually between 4% and 5% but has been observed to be as low as 1.1% or as high as 9.6% [27]. As the AD process occurs in anoxic conditions, the majority of mineralized N in FWAD is in the reduced form of $NH_4^+$, in some cases comprising over 99% of total N content [28], but typically ranges between 60% and 80% $NH_4^+$ to N [29]. High $NH_4^+$ can cause ammonium toxicity in plants and, thus, it is generally recommended to keep the proportion of total N in $NH_4^+$ and urea ($CH_4N_2O$) forms below 40% [30]. When directly applied to plants without pretreatment involving dilution or nitrification, high concentrations of $NH_4^+$ in FWAD can induce $NH_4^+$ toxicity [13,15,20,31,32], especially in hydroponic systems where nitrifying bacteria are not already present, as in many soil-based substrates [32].

High levels of $NH_4^+$ in FWAD can be remedied through nitrification processes (Table 2) [14,19]. Nitrification is facilitated by microorganisms in aerobic conditions that oxidize $NH_4^+$ to nitrite ($NO_2^-$) and, ultimately, nitrate ($NO_3^-$) [18]. Depending on the source of the organic matter used and the desired chemical composition of the FWAD, a nitrification period of up to 50 days may be required for its effective use as a fertilizer in hydroponic systems [19]. When Pelayo Lind et al. [14] applied a moving bed biofilm reactor (MBBR) technique (refer to Figure 1 in [14] for details on the MBBR technique) for the controlled nitrification of FWAD, they observed a decrease in $NH_4^+$ of 93% and an increase in $NO_3^-$ from negligible levels to almost 50% of total N content. Weimers et al. [19] reported the maximum level of the conversion of $NH_4^+$ to $NO_3^-$ occurred on day 28 while using a MBBR technique over the course of 51 days. An alternative technique to alleviate high levels of $NH_4^+$ is struvite precipitation, which uses Mg to precipitate and isolate $NH_4^+$, forming magnesium ammonium phosphate ($MgNH_4PO_4$), and precipitates due to being insoluble (Table 2) [33–35]. Ozonation is another process where ozone ($O_3$) oxidizes $NH_4^+$, leading to the formation of $NO_3^-$ (Table 2) [36,37]. This technique involves forcing $O_3$ into a FWAD solution, allowing an oxidation reaction to take place, which improves the odor of FWAD as well as decreases the ratio of $NH_4^+$ to $NO_3^-$, which provides a better source of N for plant growth [37].

**Table 2.** Suggested techniques used to reduce high levels of $NH_4^+$ in the food waste anaerobic digestates.

| Techniques | Nitrification | Struvite Precipitation | Ozonation |
|---|---|---|---|
| References | [14,19] | [34,35] | [36,37] |
| Mechanisms | Nitrification of $NH_4^+$ into $NO_3^-$ by microorganisms | Precipitation of $NH_4^+$, Mg, and $PO_4$ into $MgNH_4PO_4$ | Oxidation of $NH_4^+$ by ozone gas |
| Results | $NH_4^+\downarrow$ (by 93%) and $NO_3^-\uparrow$ by using a moving bed biofilm reactor [14]. Maximum conversion rate from $NH_4^+$ to $NO_3^-$: 11.7 g $N/m^3/d$ [19]. | $NH_4^+\downarrow$ (by 74%) at a pH of 10 [34]. $NH_4^+\downarrow$ (by 92%) at a pH of 9 [35]. | $NH_4^+\downarrow$ and $NO_3^-\uparrow$ by microbubble generator supplying 6.2 mg $O_3$/L/min for 20 min [37]. |

### 3.2. High NaCl

Common elements found in excess in FWAD include $Na^+$ and $Cl^-$, which can be attributed to the high salt content in processed food waste. The reported concentration of $Na^+$ ranges between 36 mg·$L^{-1}$ and 319 mg·$L^{-1}$, while the concentration of $Cl^-$ ranges from 87 mg·$L^{-1}$ to 2100 mg·$L^{-1}$ for FWAD [18,19,31,32]. High NaCl content in fertilizers can be detrimental to plant growth by reducing the water potential of the solution and reducing water and nutrient uptake by plants, in addition to the direct toxicity caused by excessive concentrations of $Na^+$ and $Cl^-$ ions [38]. In cucumbers (*Cucumis sativus*), a concentration of 379 mg·$L^{-1}$ and 613 mg·$L^{-1}$ for $Na^+$ and $Cl^-$, respectively, has been shown to reduce overall fruit yield in hydroponic systems [39]. Lettuce (*Lactuca sativa*) plants grown hydroponically began to show reduced fresh matter at a concentration of NaCl as low as 1167 mg·$L^{-1}$ [4,19,38,40]. To avoid the negative impacts on plant growth from high NaCl, a 20–40% dilution of FWAD is generally recommended [4,31].

### 3.3. Less Available Nutrients including P and S

The AD process causes much of the P present in the organic matter used to consolidate into the solid portion through the formation of insoluble salts [41]. The reported P concentrations in the liquid portion of FWAD range from 37 mg $L^{-1}$ to 300 mg $L^{-1}$ [18,19]. In addition, when the pH of liquid FWAD is high, a high calcium (Ca) to Mg ratio can facilitate the formation of insoluble, complex Ca-P compounds, mainly hydroxylapatite

(Hap, $Ca_5(PO_4)_2OH$), causing P to be less available for plant uptake [19,41,42]. Similarly, the AD process can also reduce the available S content in the FWAD by releasing volatile sulfur compounds, such as hydrogen sulfide ($H_2S$), and forming insoluble iron sulfides. S levels have been observed to be between 2.87 $mg \cdot L^{-1}$ and 44.23 $mg \cdot L^{-1}$ [19,43]. Low concentrations of B, Mg, Mn, and Mo have been observed as well, below the threshold typically required for many crops (Table 1). Weimers et al. [19] showed that the supplementation of P, S, Mg, Ca, Mn, B, and Mo by adding mineral salts to the liquid FWAD increased the shoot fresh mass (FM) of bok choy (*Brassica rapa* subsp. *Chinensis*).

### 3.4. pH Fluctuation

FWAD is typically alkaline with a pH > 7 as a result of $OH^-$ production when organic acids are converted to $CH_4$ during the AD process [26]. However, the pH in FWAD can be affected by the pretreatment processes. For example, during the nitrification process, the pH of FWAD can decrease dramatically; in the case of Cytryn et al. [44], a decrease in pH from 7.4 to 4.9 in 9 days was observed when nitrifying bacteria were applied to a high ammonium fertilizer. Heat treatments have been shown to influence the pH of FWAD, with Cheong et al. [43] observing an increase in pH from 7.8 to 8.7 and 9.8 when applying a 60 °C and 121 °C heat treatment, respectively. This coincided with a decrease in $NH_4^+$ concentration caused by the removal of $CO_2$ and ammonia stripping. In addition, when using FWAD in a hydroponic system, the interactions between the conversion of $NH_4^+$ to $NO_3^-$, plant N uptake, and N availability can cause pH fluctuation in the nutrient solution during plant growth. Pelayo Lind et al. [14] observed a decrease in pH from 6.6 to 4.5 when using non-nitrified FWAD after 15 days of bok choy growth from seedlings. This decrease in pH was attributed to a high presence of $NH_4^+$ and high plant $NH_4^+$ uptake. In contrast, applying nitrified FWAD with facilitated nitrification using nitrification reactors caused an increase in pH from 5.3 to 6.6 associated with increases in $NO_3^-$ content and plant $NO_3^-$ uptake. Van Rooyen and Nicol [45] also demonstrated that in the presence of ammonia oxidizing bacteria, the conversion of $NH_4^+$ to $NO_3^-$ induces an acidic effect due to the oxidation of $NH_4^+$, and it was reported that $NH_4^+$ oxidation was being carried out by bacteria ten times faster than the $NH_4^+$ absorption rate by plants alone. Using this relationship, it has been suggested to add $NH_4^+$ in the form of FWAD when the nutrient solution becomes basic, or to add KOH or $NO_3^-$ using the nitrified FWAD when the nutrient solution has an acidic pH [14,45].

### 4. Effects of Using FWAD on Crop Production

FWAD has great potential as a fertilizer, but using FWAD as a primary nutrient source can be challenging as application methods have yet to be established. In addition, the crops must perform at least as well as conventionally grown plants with other synthetic and organic fertilizers to ensure that FWAD can be adopted by growers for large-scale applications. Here, we summarize recent studies that compared crop growth under FWAD with those grown using synthetic fertilizers (Table 3). In addition, we reviewed studies that investigated how the application of FWAD influences crop growth under different dilution concentrations, nitrification control, and the use of biochar (Table 3).

**Table 3.** Case studies testing food waste anaerobic digestate (FWAD) fertilizers treatments with comparable fertilizer treatments (↑: increase, ~: similar, ↓: decrease in plant responses under FWAD treatments compared to under control).

| Control | FWAD Treatments | Crop | Crop Responses | Reference |
|---|---|---|---|---|
| Synthetic fertilizer at 60 ppm nitrogen (N) during vegetative growth and 100 ppm N during fruiting | FWAD at 60 ppm N during vegetative growth and 100 ppm N during fruiting | Tomato | Survival rate ↓ Plant height ~ | [32] |
| Synthetic fertilizer at electrical conductivity (EC) of 0.5 mS·cm$^{-1}$ | FWAD at EC of 0.5–8.5 mS·cm$^{-1}$ | Lettuce | At EC of 0.5 mS·cm$^{-1}$: Fresh yields ~ At EC of 4.5, 6.5, 8.5 mS·cm$^{-1}$: Fresh yields ↑ | [12] |
| Synthetic fertilizer at EC of 0.5 mS·cm$^{-1}$ | FWAD at EC of 0.5–8.5 mS·cm$^{-1}$ | Parsley | At EC of 0.5 mS·cm$^{-1}$: Fresh yields ~ At EC of 2.5, 3.5 mS·cm$^{-1}$: Fresh yields ↑ | [12] |
| Synthetic fertilizer at EC of 2.3 mS·cm$^{-1}$ | FWAD at EC of 2.3 mS·cm$^{-1}$ | Tomato | Fresh fruit yield ↓ Number of fruits ~ Fruit dry matter content ~ | [20] |
| Synthetic fertilizer amended to a similar nutrient concentration as FWAD at 250 ppm N | Nitrified FWAD at 250 ppm N | Bok choy | Shoot fresh mass ~ Shoot dry mass ~ Chlorophyll content ~ | [19] |
| Synthetic fertilizer at 250 ppm N | Nitrified FWAD amended to a similar nutrient concentration as synthetic fertilizer at 250 ppm N | Bok choy | Shoot fresh mass ~ Shoot dry mass ↑ | [19] |
| Synthetic fertilizer at 720 ppm N | FWAD at a concentration of 20%, 40%, 60%, 80%, 100% | Bok choy | At 20–80%: Fresh mass ~ Dry mass ~ At 100%: Fresh mass ↓ Dry mass ↓ | [43] |
| Synthetic fertilizer | FWAD at a concentration of 20%, 40%, 60%, 80%, 100% | Chinese spinach, water spinach, bok choy, lettuce | At 20–40%: Shoot fresh mass ~ | [4] |
| Synthetic fertilizer at 192 ppm N | FWAD (at 210 ppm inorganic N) with a facilitated nitrification process, Nitrified FWAD (at 182 ppm inorganic N) without or with facilitated nitrification process | Bok choy | Shoot fresh mass ~ | [14] |
| FWAD at 0, 2, 4, 6, 8, 10% without biochar | FWAD at 0, 2, 4, 6, 8, 10% with biochar | Tomato | At 2, 4, 6%: Shoot dry mass ↓ Shoot N uptake ↓ | [18] |
| Synthetic fertilizer | FWAD amended biochar | Lettuce, kale, rocket salad | Shoot fresh mass ~ | [4] |

### 4.1. Comparisons with Synthetic Fertilizer

Nitrogen is often the most limiting nutrient for plant growth, and the fertilizer concentration is determined to meet the plant demand for nitrogen. A comparison between tomato plants (*Solanum lycopersicum*) fertilized with FWAD and those fertilized with synthetic fertilizer was performed at the same nitrogen level in a deep-water-culture hydroponic system which was set up outdoors in a full-sun area [32]. FWAD and a 15N-2.2P-11.6K synthetic fertilizer were initially applied at 60 ppm total N and raised to 100 ppm once fruit began to develop. In the synthetic fertilizer and FWAD at 60 ppm total N, there was a $NO_3^-$ concentration of 52 ppm and <1 ppm, and an $NH_4^+$ concentration of 12 and 32 ppm, respectively. The pHs of the initial nutrient solution made with synthetic fertilizer and FWAD were 6.95 and 8.31, respectively, and a slight increase in pH occurred over the course of the experiment for both treatments. Although a 75% loss in tomato plants was observed in the plants given FWAD, the surviving tomato plant had a similar plant height to those given synthetic fertilizer. The loss in tomato plants was primarily attributed to $NH_4^+$ toxicity, which caused a stunted root system to develop for the plants given FWAD. The high pH of the FWAD fertilizer could have also caused nutrients to be unavailable for plant uptake. It was initially expected that nitrification could take place over the course of the experiment without pretreatment being required, but the high levels of $NH_4^+$ and low levels of $NO_3^-$ suggested that effective nitrification did not take place.

Electrical conductivity (EC) measures the total concentration of fertilizer salts in a solution and is sometimes used to manage nutrient concentrations. Stoknes et al. [12] investigated how varying EC from 0.5 to 8.5 mS·cm$^{-1}$ in FWAD influenced the growth of lettuce and parsley (*Petroselinum crispum*) grown on peat substrate, compared to a synthetic fertilizer at an EC of 0.5 mS·cm$^{-1}$ in an insulated greenhouse with natural lighting. At the same EC of 0.5 mS·cm$^{-1}$, FWAD and a synthetic fertilizer produced similar fresh yields. An EC of 4.5, 6.5, and 8.5 mS·cm$^{-1}$ in the FWAD increased lettuce yield by 163%, 150%, and 156%, respectively, compared with a synthetic fertilizer at an EC of 0.5 mS·cm$^{-1}$. A 93% and 113% increase in yield in parsley plants was also observed at ECs of 2.5 and 3.5 mS·cm$^{-1}$, respectively, compared with a synthetic fertilizer at an EC of 0.5 mS·cm$^{-1}$. In a separate experiment growing tomatoes under similar conditions, Stoknes et al. [20] diluted FWAD to have a comparable EC with a synthetic fertilizer control of 2.3 mS·cm$^{-1}$. During the experiment, higher pH and Cl$^-$ levels, and lower dissolved oxygen and P, were observed in the nutrient solution made with FWAD when compared to a synthetic fertilizer. FWAD produced 25% lower fresh matter fruit yield per container than a synthetic fertilizer, while FWAD and a synthetic fertilizer had a similar number of fruits per container and dry matter of fruits.

The nutrient characteristics of FWAD differ from synthetic fertilizers in terms of the composition and concentration of macro- and micronutrients. Weimers et al. [19] compared the growth of bok choy grown hydroponically, using a peat substrate in a greenhouse under natural light, using synthetic fertilizer and FWAD that have similar macro- and micronutrient concentrations. Each fertilizer treatment was diluted to a concentration of 250 ppm total N. For one comparison, the synthetic fertilizer was adjusted so that it would resemble the nutrient composition of the FWAD. In another comparison, FWAD was supplemented with Mg, Ca, P, Mn, B, and Mo so that it more closely resembled the mineral content of a standard synthetic fertilizer. The non-supplemented FWAD, and the synthetic fertilizer formulated to match it, had similar effects on plant growth, producing similar dry mass (DM) and FM yields and chlorophyll content. However, the modified FWAD treatment achieved a 17% higher DM yield than its mineral equivalent. A significant decrease was initially observed between P uptake in the unmodified FWAD and its mineral equivalent, with 65% and 83% P uptake observed, respectively. When supplemented with additional P in the modified FWAD treatment, the P use efficiency was similar in the modified FWAD and synthetic fertilizer. The low phosphorus use efficiency values in the FWAD were believed to be due to a high Ca-to-Mg ratio, which can favor the formation of insoluble, complex Ca-P compounds. Another possibility was the high Fe-to-Mg ratio,

which can cause less P to be available for plant uptake. A low level of S was also observed in plant tissues using the unmodified FWAD, with 71% of total S use efficiency when compared to its mineral equivalent, which was believed to be due to the formation of insoluble iron sulfides and a higher N-to-S ratio that resulted in less total S being applied when formulating for N concentration.

### 4.2. Effects of Concentrations of Food Waste Anaerobic Digestate

FWAD has a high concentration of $NH_4^+$ and salts ($Na^+$ and $Cl^-$). The dilution of FWAD to lower these concentrations has been suggested as one of the simplest ways to reduce issues of $NH_4^+$ and salt toxicity. Cheong et al. [43] investigated the effect of the dilution of three types of FWAD (untreated or heat-treated at 60 °C and 121 °C) at a concentration of 20%, 40%, 60%, 80%, and 100% (undiluted) compared to a 15N-6.6P-12.5K fertilizer at 720 ppm N. Bok choy was grown in pots using cocopeat and biochar as substrates, and plants were grown at the National University of Singapore Native Plant Nursery using natural light. The total ammonia nitrogen of untreated, 60 °C-treated, and 121 °C-treated FWAD was 4700 ppm, 4500 ppm, and 3100 ppm, respectively. Regardless of heat treatments, shoot FM and DM were similar between the FWAD treatments and the commercial synthetic fertilizer treatment, except at a 100% concentration of FWAD where shoot FM and DM decreased by at least 50% in each crop tested. In the following study [4], the concentrations of 20%, 40%, 60%, 80%, and 100% of untreated FWAD were tested against controls of tap water and a 15N-6.6P-12.5K fertilizer on a variety of leafy vegetables, including Chinese spinach (*Amaranthus tricolor*), water spinach (*Ipomoea aquatica*), bok choy, and lettuce, in similar growing conditions to the previous study. For all species, a concentration of 20–40% FWAD produced a similar shoot FM compared to the synthetic fertilizer. In both studies [4,43], the poor results in shoot FM at higher FWAD concentrations were attributed to the high salinity, high $NH_4^+$ concentration, or/and low oxygen level in the rootzone associated with the high chemical oxygen demand of FWAD with little dilution applied.

### 4.3. Effects of Controlling Nitrification

When FWAD is used in hydroponic systems, in addition to high $NH_4^+$, the pH can be particularly difficult to manage because of the effects of nitrification as well as the uptake of $NH_3$, $NH_4^+$, and $NO_3^-$ by plants. Pelayo Lind et al. [14] investigated how using pre-nitrified FWAD and/or facilitating nitrification using nitrification reactors (see Figure 1 of [14]) during cultivation influences the pH dynamics in the nutrient solution and plant growth of bok choy in a greenhouse using a hydroponic nutrient film technique with supplemental lighting from high-pressure sodium lamps. Pre-nitrification increased the $NO_3^-$ portion of the total inorganic N from <1% to 50% while decreasing the ammonium portion from >99% to 8% and pH from 8.2 to 5.0. The total inorganic N was reduced from 210 ppm to 182 ppm after nitrification. Over time, a decrease in the pH of the nutrient solution was observed when non-nitrified FWAD was used due to a higher concentration of $NH_4^+$, resulting in greater $NH_4^+$ uptake by plants. In contrast, using nitrified FWAD with a facilitated nitrification process together increased pH, likely due to increased levels of $NO_3^-$ being taken up by the plants. Facilitating nitrification by adding non-nitrified FWAD when the pH dropped below 5.8 was the only treatment that stabilized the pH at around 5.8. When the non-nitrified FWAD was added to the system, as the $NH_4^+$ levels decreased and the $NO_3^-$ levels increased via the nitrification process, the pH increased. However, large amounts of FWAD were required for this to take place (1.5 L of diluted FWAD per day was used), which can be problematic because the addition of large amounts of FWAD, even if it has been nitrified, can lead to high levels of $NH_4^+$ and salts, leading to toxicity in plants and poorer access to oxygen for plant roots due to the high chemical oxygen demand in the FWAD. Three weeks after transplanting, the shoot FM and DM of bok choy were similar between the nitrified and non-nitrified FWAD in the system. However, the shoot FM (by 50–69%) and DM (by 42–66%) were smaller under FWAD treatments than under

the synthetic fertilizer control with 192 ppm N, except for the shoot DM which was similar under pH-controlled FWAD and synthetic fertilizer. The shoot FM and DM of bok choy grown under FWAD for four weeks were similar to those grown with synthetic fertilizer for three weeks.

### 4.4. Effects of Using Food Waste Anaerobic Digestate with Biochar

Biochar is a solid substance obtained by heating organic biomasses, such as plant matter or agricultural residues, in an anaerobic environment [17,46,47]. After processing, biochar has a high surface area, increasing microbial activity and water retention, as well as reducing the leaching of nutrients, and ultimately improving plant growth when used as an amendment for substrates [48]. It can also reduce the susceptibility of plants to salt stress by decreasing $Na^+$ uptake [49]. In addition, biochar can impact the nitrification process by adsorbing $NH_4^+$ and altering the presence of ammonia-oxidizing bacteria [46]. Mickan et al. [18] investigated how the addition of biochar to potting mixes affects the use of FWAD on plant growth of tomatoes in a glasshouse. The FWAD used in the study had a 5000 ppm of total N and was applied with 0, 2, 4, 6, 8, and 10% of the total potting mix volume. When biochar was present, 34% more FWAD was required for plants to reach a threshold of 95% maximum shoot DM when compared with the control of no addition of biochar. Similarly, for equal levels of shoot N uptake to be observed, higher concentrations of FWAD were required with the biochar treatment than when FWAD was added alone. An inhibitory relationship was observed between the addition of biochar and the nitrification process, resulting in less conversion of $NH_4^+$ to $NO_3^-$ which in part explains the poor N availability and plant growth with biochar. These results suggest that with excess FWAD needing to be disposed of from biogas plants, the addition of biochar to substrates could allow more FWAD to be applied to plants, leading to less waste disposal. In addition, the effects of lower N uptake in substrates amended with biochar could allow FWAD to be used in a similar way to other slow-release organic fertilizers.

Biochars can increase the long-term availability of nutrients to plants by absorbing nutrients and slowly releasing them. Song et al. [4] tested whether two types of wood-based biochar that were used to filter 4700 ppm total ammonia nitrogen FWAD could be used as a fertilizer source to grow lettuce, kale (*Brassica oleracea* var. *sabellica*), and rocket salad (*Eruca sativa*) under natural light. In all three plants studied, a similar shoot FM was observed between the FWAD-amended biochar and a 15N-6.6P-12.5K synthetic fertilizer control. In addition, when the biochar amended with FWAD was compared with biochars sourced from black soldier fly larval frass or municipal waste, the shoot FM of lettuce, kale, and rocket salad was greater with the FWAD-amended biochar than with either of the other biochars. Combining biochar and digestate has been shown to confer benefits to plant fertilization including a more stable pH, an improved electron transfer, and an improvement in microbial communities [50]. The biochar sourced from black soldier fly larval frass had a pH of above 8, and both the black soldier fly larval frass and municipal waste biochar had excess levels of $Ca^{2+}$, which may have negatively impacted growth. The FWAD-amended biochar did not exhibit these poor qualities to nearly the same extent, suggesting that this type of biochar could be more favorable for use in plant growth and create a more sustainable process for the dilution and use of FWAD.

### 5. Conclusions

The use of FWAD as fertilizer has the potential to increase the sustainability of crop production on a variety of fronts. While reducing waste and greenhouse gas emissions, FWAD contains a high concentration of N, typically in the form of $NH_4^+$. The low levels of food recycling result in a great supply of inputs into a potential circular economy approach to the food cycle. For crop production, the use of FWAD can increase yield and help facilitate a more sustainable process. However, there are still challenges associated with the effective use of FWAD for crop production. In order for FWAD to be used as an effective fertilizer, pretreatment is recommended to convert $NH_4^+$ to $NO_3^-$ using facilitated nitrification,

and dilution is effective for reducing the concentration of salts that could otherwise be at levels that are too high for effective plant growth. Amending FWAD with mineral components also improves nutrient balances and fertilizer efficacy. Several industrial technologies (such as ammonia stripping, adsorption, membrane filtration/concentration, struvite crystallization/precipitation, and evaporation) are also suggested for efficient nutrient recovery for effective fertilization using FWAD [51]. Future research should focus on the further optimization of techniques, as well as on new processes that help address the current challenges of using FWAD as a fertilizer.

**Author Contributions:** Conceptualization, J.R., Z.C. and Y.P.; methodology, J.R. and Y.P.; validation, J.R., Z.C. and Y.P.; writing—original draft preparation, J.R. and Y.P.; writing—review and editing, J.R., Z.C. and Y.P.; supervision, Y.P.; project administration, Y.P.; funding acquisition, Y.P. and Z.C. All authors have read and agreed to the published version of the manuscript.

**Funding:** The APC was funded by the Zimin Institute for Smart and Sustainable Cities at Arizona State University.

**Data Availability Statement:** No new data were created or analyzed in this study. Data sharing is not applicable to this article.

**Conflicts of Interest:** The authors declare no conflict of interest. The funders had no role in the design of the study; in the collection, analyses, or interpretation of data; in the writing of the manuscript; or in the decision to publish the results.

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
