# Peer review of "Potential Applications of Food-Waste-Based Anaerobic Digestate for Sustainable Crop Production Practice"

_sustainability, doi:10.3390/su15118520_

Round 1

Reviewer 1 Report

The article provides a comprehensive overview of the use of food waste anaerobic digestate (FWAD) as a fertilizer for crop production. The authors have presented a well-structured and organized manuscript that covers a range of topics related to FWAD, including its composition, properties, and effects on plant growth. While the article provides a good overview of the topic, there are a few areas that could be improved as listed below.

1. One potential issue is that the article could benefit from more specific examples and case studies to illustrate the challenges and benefits of using FWAD in crop production.

2. Furthermore, while the authors did discuss some of the challenges associated with using FWAD, such as the need for pretreatment and dilution, it would have been beneficial to see more discussion on potential solutions to these challenges. For instance, what new processes or techniques are being developed to address these issues?

3. Finally, the article could benefit from a more thorough discussion of the potential environmental impacts of using FWAD as a fertilizer, particularly in terms of nutrient runoff and its effects on water quality.

Overall, the article is well-written and informative, and the authors provide a strong foundation for future research in the field. However, some additional information and discussion could have strengthened the manuscript even further.

Reviewer 2 Report

I reviewed the manuscript entitled "  Potential Applications of Food Waste Based Anaerobic Diges-2 tate for Sustainable Crop Production Practice ".  The manuscript has a lot of information. The manuscript needs some correction before it can be published. I suggest authors take a closer look and adjust the write up to be more precise and appealing to the readers.

1.     Please improved the abstract. The abstract should have one sentence per each: context and background, motivation, hypothesis, methods, results, conclusions. At the end of the abstract section, the conclusion of the work should be stated in one sentence.

2.     Line sizes are not the same on some pages. Please correct it.

3.     The background of the relevant research should be given in the introduction section.

4.     The purpose and innovation of the research should be clearly stated at the end of the introduction section.

5.     Remove vertical lines in tables.

6.     If some results are quantitatively expressed, the quality of the article will be better.

7.     The sources are completely up-to-date and appropriate.

8.     The English are generally OK.

9.     The explanations of each of the parameters are very good.

Reviewer 3 Report

A very good and well-written review paper. Just two corrections are needed in lines 290 and 359, respectively.

Reviewer 4 Report

The manuscript well summarized the properties of FWAD, the effect of FWAD on crop production, and  the challenges associated with the effective use of FWAD for crop production. The article is interesting and in good writing style. Please check it and correct minor mistakes in grammar and spelling.

Reviewer 5 Report

Reviewer comments

Article: Potential Applications of Food Waste-Based Anaerobic Digestate for Sustainable Crop Production Practice

Authors:  Jonathan Ries, Zhihao Chen and Yujin Park

The manuscript entitled “Potential Application of Food Waste Based Anaerobic Digestate for Sustainable Crop Production Practice contributes more information and update on the nutritional properties of FWAD and recent research employing it in crop and horticultural production to deepen understanding of the benefits and difficulties of FWAD use as a fertilizer. However, the following points need to be evaluated before accepting the manuscript.

Note: In the manuscript, line no. is highlighted and the proper comment information is given below

Line 52-54: In food waste the C: N ratio is less. Due to the high amount of nitrogen in food waste, there is an emission of GHGs in this too. Then how do you defend against it? Explain proper justification.

Line 60-62: You have given more advantages of anaerobic digestate in your review. But there are many disadvantages too.  Like in anaerobic digestate, there are some bacteria and fungi which can cause disease in the plant. Then how can we use it in fertilizer...? Please justify the explanation. (For more information you can refer the following paper: https://doi.org/10.1016/j.envres.2023.115529)

Line 67: Explain Table 1 well. What do you want to tell in Table 1, it is not understood.

Line 92: A lot of information is available in this section. Make this section more informative.

Line 99-100: Liquid digestate will also contain some bacteria and fungi which will cause disease in plants. Digestion will take the process of fermentation in the hydroponics technique too. Because digestate consists of many microorganisms. Please explain the proper justification behind that or explain the effective technology used by this system.

Line 134: Give the mechanism of the nitrification process.

Line 139: Explain this technique nicely by making a figure.

Line 140-150: You can make a good table of this technique and its uses, significance, and mechanism.

Line 166: Why these nutrients are less in AD? Give a justification.

Line 182: Give justification

Line 197: Reference….?

Line 215: In this section, you can make a table format with additional previous research. In this section, many references are missing. Please add missing references in the manuscript.

Line 220-221: References missing.

Line 226-227: References missing.

Line 247-249: References missing.

Line 290: Botanical name always in italics format.

Line 298: Many references are missing. Please add missing references in the manuscript.

Line 304-304: Draw a figure of this technique or add an image of it.

Line 328: References Missing

Line 333-334: References Missing

Line 349: References Missing

Line 355-357: Give justification behind that. (Please refer to this paper. https://doi.org/10.1016/j.jclepro.2021.127143)
